# Learning Reactive Synthesis from Model Checking Feedback

## Abstract

Deep learning applications to formal verification typically fall into one of two categories: employing reinforcement learning that suffers from slow convergence, or supervised learning that suffers from limited exploration. For reactive synthesis, the problem of automatically constructing a system that satisfies a formal specification, existing approaches fall into the latter category. In this paper, we propose a hybrid approach that only initializes the model with supervised learning and then continues training by reinforcing formally verified predictions. We show that by training the model to synthesize correct solutions rather than fixating on the supervised data, performance substantially improves. We can further utilize our approach to optimize for size without any performance degradation. Finally, we show that we can iteratively reinforce on open problems that synthesis tools are unable to solve. Our approach is demonstrated for both deep neural networks trained from scratch and pre-trained models fine-tuned on reactive synthesis, establishing new state-of-the-art results for learning reactive synthesis.

## 1 Introduction

Reactive synthesis is one of the fundamental problems in formal verification: Given a specification in a formal logic, a synthesis algorithm automatically constructs a system that satisfies the specification (Church, 1963). The promise of making the manual implementation of systems superfluous has sparked research for more than half a century, ranging from early theoretical contributions (Buchi & Landweber, 1969) to modern algorithms and tools (Meyer et al., 2018; Renkin et al., 2022; Kretínský et al., 2025). Nowadays, an annually held competition tracks the progress in the field (Jacobs et al., 2022). Most research interest revolves around reactive synthesis from specifications expressed in linear-time temporal logic (LTL) (Pnueli, 1977) for which the synthesis results are hardware circuits. The success story of LTL started in hardware model-checking, where a multitude of industry-level model-checkers have been developed Kuppe et al. (2019), which eventually led to a widely applied industry standard (IEEE-Commission, 2005). The computationally harder synthesis problem from LTL specifications is theoretically complex and often intractable in practice, as existing synthesis algorithms often time out even for small specifications. One promising way to overcome these barriers is deep learning: in recent years, reactive synthesis was turned into a deep learning problem (Schmitt et al., 2021) to build on the promising success of deep learning in program synthesis (Austin et al., 2021) and code generation (Rozière et al., 2023).

Current deep learning solutions, however, suffer from the drawbacks of supervised learning: They generalize over the training data and mimic the behavior of the synthesis tool used for data generation. To enable supervised learning, large datasets of specification-circuit pairs are constructed with the help of algorithmic synthesis tools for synthesizing the training targets (Schmitt et al., 2021). While supervised learning on such datasets allows for some generalization, the resulting systems are ultimately confined to the abilities of the synthesis tool that generated the training data. In the reinforcement learning literature, such problem settings are regarded as imitation learning or learning from demonstration (Schaal, 1996), i.e., training an agent to imitate a teacher that is providing labeled data. In the context of synthesis, the learning algorithm can be seen as imitating the specific synthesis algorithm and tool that was used to generate the dataset. Imitation learning is well-known to have limited ability to fully generalize or substantially improve over the labeled data (Ross & Bagnell, 2010). Therefore, it appears unlikely that a fully supervised learning approach alone will overcome the limitations of reactive synthesis tools.

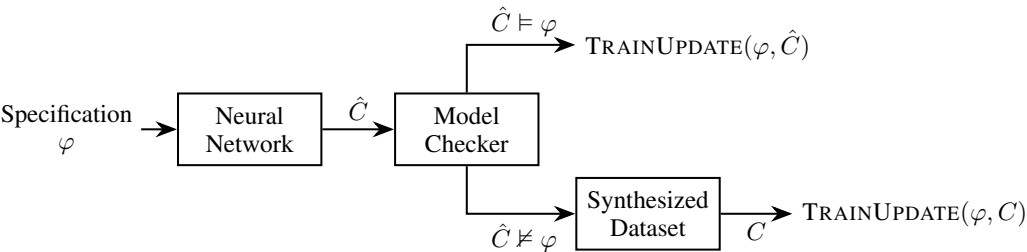

Figure 1: An overview of our method showing the conditional training update. For a given specification $\varphi$, the neural network predicts a circuit $\hat{C}$ that is verified against $\varphi$ by a model-checker. In case $\hat{C}$ satisfies the specification ($\hat{C} \vDash \varphi$), we perform a gradient update with $\varphi$ and $\hat{C}$. In the dual case $\hat{C} \nvDash \varphi$, a synthesis tool is used to compute a correct circuit $C$ for the gradient update.

In this paper, we present a new learning approach for reactive synthesis that overcomes the restrictions of imitation learning. Our approach uses supervised learning only as an initialization of the neural network, after which a second training stage is entered that allows the neural network to self-improve its own predictions. Depicted in Figure 1, we formally verify the circuits $\hat{C}$ that was predicted by the neural network and utilize this feedback to train on circuits that the neural network already predicts correctly. If the prediction was correct, we reinforce the model with a training update on the specification and its prediction. We only fall back to a training target $C$ from a synthesis tool if the neural network is not yet able to predict a correct circuit ($\hat{C} \nvDash \varphi$). The formal verification of the prediction is achieved by utilizing existing model-checking tools and benefits from the lower complexity of the model-checking problem for LTL. We thereby change the training objective from imitating a synthesis tool to the actual objective, i.e., the synthesis of a circuit that satisfies the specification. The effects of this change are substantial, as shown in our experiments. Our approach not only improves sample efficiency but also allows our model to generalize better. We can amplify the results by searching over multiple predictions of the model and make use of it for optimizing the size of circuits. Finally, we show that our method clearly scales beyond the capabilities of the synthesis tool used to generate the data by including open problems into the training process.

In summary, we make the following contributions:

1. We introduce a novel deep learning approach to reactive synthesis, combining both supervised and reinforcement learning to optimize correctness over imitating synthesis tools.

2. We generalize our approach with a search component to an expert iteration method and demonstrate its ability to further improve performance and optimize the size of circuits.

3. We utilize our framework to iterate on synthesis problems that reactive synthesis tools are unable to solve and show the potential of our methods to gradually improve on them.

The remainder of the paper is structured as follows. Related work is discussed in Section 2. Background on reactive synthesis and deep learning methods thereof are described in Section 3. Our method for learning reactive synthesis from model checking feedback is presented in Section 4 followed by an experimental evaluation in Section 5. We conclude and discuss future work in Section 6.

## 2 RELATED WORK

**Expert Iteration and Theorem Proving.** Expert iteration (Anthony et al., 2017) has been applied with great success to automated theorem proving. An early example in the context of large language models is the GPT-f work by Polu & Sutskever (2020) which was later developed into a full curriculum learning approach by Polu et al. (2023). Recent applications to Lean corpora such as the Lean-workbook corpus (Ying et al., 2024) were presented by Wu et al. (2024) and Xin et al. (2025).

**Deep Learning Aided Verification and Synthesis.** The importance of formal methods in domains such as hardware design has led to extensive application of deep learning to verification and

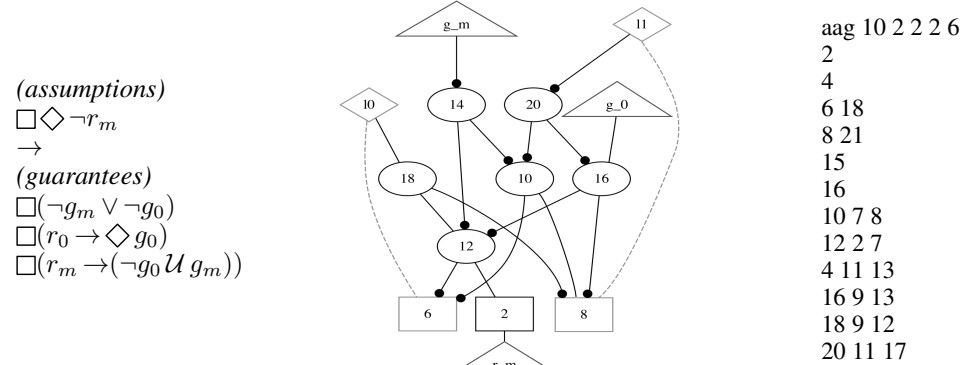

Figure 2: We show an example for an LTL specification that specifies a prioritized arbiter on the left. A circuit implementing the specification is provided as a graphical representation (middle) and in the AIGER format (right).

synthesis tasks, covering all steps of the verification process. Beginning with formal specifications, deep learning was used to automatically formalize natural language into specifications (Chen et al., 2023; Cosler et al., 2023a; Mendoza et al., 2024). For the verification of formal specifications neural networks were used as proof certificates (Giacobbe et al., 2024). Deep learning has been applied to hardware systems themselves at various levels of abstraction, including boolean circuits (Neto et al., 2021; Chowdhury et al., 2024; Wang et al., 2024), hardware description languages (Vasudevan et al., 2021; Thakur et al., 2024; Zhu et al., 2022; Zheng et al., 2025) and chip placement (Mirhoseini et al., 2021). For reactive synthesis specifically, a neural-symbolic portfolio solver was developed (Cosler et al., 2024) following the work of Schmitt et al. (2021). The same architecture that was applied to reactive synthesis was also applied to the related problem of circuit repair (Cosler et al., 2023b).

**Deep Learning for Formal Logics.** Deep learning has proven itself to be a promising direction for solving formal logic problems. For Boolean Satisfiability (SAT) supervised (Selsam & Bjørner, 2019; Selsam et al., 2019; Li et al., 2024), unsupervised (Amizadeh et al., 2019; Ozolins et al., 2022) and expert iteration approaches (Ghanem et al., 2024) were explored for both predicting and proving satisfiability. For quantified Boolean formulas (QBF) (Lederman et al., 2020) and Satisfiability Modulo Theories (SMT) (Balunovic et al., 2018) deep neural networks were successfully integrated into algorithmic solvers. For temporal logics such as LTL, most research focused on learning traces for satisfiability prediction (Hahn et al., 2021; Luo et al., 2022; Isik et al., 2024). Closely related to temporal logics, representation of Büchi automata were learned with Graph Neural Networks (Stammet et al., 2022).

## 3  BACKGROUND

Reactive synthesis is a fundamental problem in computer science (Church, 1963), with theoretical solutions already established in the 1960s (Buchi & Landweber, 1969). The most common version is reactive synthesis, where a circuit is synthesized for a provided temporal, e.g., linear-time temporal logic (LTL) (Pnueli, 1977). LTL combines propositional boolean logic operators such as $\neg, \wedge, \vee, \implies$ with temporal operators such as $\bigcirc$ - *next*, $\mathcal{U}$ - *until*, $\square$ - *always*, and $\diamondsuit$ - *eventually* which allows to specify the behavior of a reactive system that maintains a continuous interaction with its environment. We give an example of an LTL specification for a prioritized arbiter in Figure 2. It specifies that, under the assumption that $\neg r_m$ is infinitely often true, $g_m$ and $g_0$ are never true at the same time, whenever $r_0$ holds, then at some later point $g_0$ holds, and that whenever $r_m$ holds, $g_0$ does not hold until $g_m$ holds. A system that is implementing such a specification is typically represented as a sequential circuit that translates an infinite stream of inputs into an infinite stream of outputs. A standard representation for sequential circuits is And-Inverter Graphs extended with simple memory elements, so-called latches. The AIGER format (Brummayer et al., 2007), which we employ in this work, is a widely adopted textual encoding for such And-Inverter Graphs. In Figure 2, we show a graphical representation of an And-Inverter Graph and its corresponding

AIGER format. The graphical representation is read from bottom to top, with triangles being the input and outputs. Diamonds are latches that store their input value for one clock cycle, and ellipses describe AND-gates. Negations are depicted by a dot on the wire between gates. Algorithms for solving LTL synthesis can be broadly categorized into game-based (Rabin, 1972) and bounded synthesis (Schewe & Finkbeiner, 2007). All algorithms face the challenges related to the problem being 2EXPTIME-complete (Pnueli & Rosner, 1989).

The complexity bound of the algorithmic approach to the problem of reactive synthesis motivated the use of machine learning methods. Supervised learning has recently been applied to reactive synthesis (Schmitt et al., 2021). To enable their approach, Schmitt et al. (2021) proposed a data generation method that leverages the synthesis tool Strix (Meyer et al., 2018) to generate large numbers of synthetic specification-circuit pairs. Since reactive synthesis specifications typically consist of conjunctions of smaller assumption and guarantee properties (see Figure 2), the authors collected a set of such properties from the annual synthesis competition (Jacobs et al., 2022) and randomly combined them to form new specifications. The generation process begins with a single property and incrementally adds new properties until either the specification becomes unrealizable or the synthesis tool times out after 120 seconds. Using this approach, the authors constructed a dataset containing hundreds of thousands of training samples. The reactive synthesis problem is then phrased as a sequence-to-sequence learning problem. The authors demonstrated that a hierarchical transformer architecture can be successfully trained on the synthetic dataset and is able to generalize to both the synthesis competition benchmarks and specifications that the synthesis tool cannot solve. The same dataset was later used to fine-tune code generation models on the reactive synthesis task, which exhibited superior generalization compared to the hierarchical transformer (Schmitt et al., 2023). We will adopt the same architectures and datasets to evaluate our approach, as detailed in Section 5.

## 4 METHOD

In the following, we describe our approach for combining supervised learning with model-checking feedback for reactive synthesis. We begin by describing the general idea and present three adapted versions in the subsequent paragraphs. Our method requires a dataset $D = \{(\varphi_1, C_1), \ldots, (\varphi_n, C_n)\}$ of specification-circuit pairs and access to a model-checking tool to automatically verify a circuit $C$ against a specification $\varphi$. Given dataset $D$, we start by building an initial model $M_0$ with standard supervised learning. Assuming that the labels in dataset $D$ were generated with a synthesis tool, we can view this first stage as an imitation learning phase that trains the model to imitate the synthesis tool. Following the initialization with imitation learning, we begin to iteratively improve our model with its own predictions. We distinguish between using top-1 and top-k predictions, as well as access to the circuit from the dataset that we can fall back to. For simplicity, we describe the methods for a single specification-circuit pair. It is straightforward to generalize this to the mini-batched algorithm that we have implemented in our experiments.

**Reinforcing Learned Semantics.** Our first method is motivated by the observation that some model predictions differ from the training targets but still satisfy the input specification. This is an expected situation since, in theory, there exist infinitely many correct circuits for each specification. A practical reason can, for example, be as simple as changing the order of gates or as complicated as creating a completely different circuit. We propose reinforcing such predictions as described in Algorithm 1. In each iteration, we sample a specification-circuit pair $(\varphi, C)$ from training data $D$ and evaluate model $M_{i-1}$ from the previous iteration on specification $\varphi$. If prediction $\hat{C}$ of our model satisfies the specification (denoted as $\hat{C} \models \varphi$), we train on $\hat{C}$ instead; otherwise, we keep the training target $C$.

**Expert Iteration and Circuit Minimization.** The previous method can be generalized by performing a search over the model outputs, evaluating the top-k circuit predictions, and selecting the training target among them. Specifically, we generalize lines 4 and 5 in Algorithm 1 by first obtaining the top-k circuits $\hat{C}_1, \ldots, \hat{C}_k$, verifying them, and then selecting our training target from the circuits satisfying the specification and the circuit $C$ from the training data. The generalized version is given as Algorithm 3 in Appendix C.

---

**Algorithm 1** Learning from Model Checking Feedback

---

**Require:** $D = \{(\varphi_1, C_1), \ldots, (\varphi_n, C_n)\}$
1: $M_0 \leftarrow \text{IMITATIONLEARNING}(D)$
2: **for** $i \leftarrow 1$ **to** $n$ **do**
3:     Sample $(\varphi, C) \sim D$
4:     $\hat{C} \leftarrow \text{EVALUATE}(M_{i-1}, \varphi)$
5:     $C^* \leftarrow$ **if** $\hat{C} \models \varphi$ **then** $\hat{C}$ **else** $C$
6:     $M_i \leftarrow \text{TRAINUPDATE}(M_{i-1}, (\varphi, C^*))$

---

This is motivated by the observation that generating more than a single output (for example, through beam search) increases the likelihood of finding correct solutions. We note that this resembles the expert iteration method proposed in the context of reinforcement learning (Anthony et al., 2017). In our context, building an expert corresponds to automatically verifying the top-k model predictions with the fallback mechanism to the training dataset. For larger $k$, it becomes likely to find multiple valid circuits, therefore raising the question of how to best choose between them. In addition to selecting the first circuit that satisfies the specification, our method is also used to optimize the size of circuits. Smaller circuits are generally preferable as they are easier to understand and, in principle, less expensive to manufacture. By selecting the smallest circuit, we are not only reinforcing the semantics but also training the model to find minimal circuits, further disentangling the model from the initial dataset.

**Iterating on Open Problems.** Finally, we consider a version of our method in which we do not always have access to a training target for a given specification. We refer to such problems as *open problems*. This is particularly interesting for improving over a synthesis tool where the open problems correspond to those that the synthesis tool is unable to solve. Note that synthesis tools run into timeouts even for specifications for which small solutions exist if they are hard to compute. We modify our method as follows: With probability $p_{open}$, we sample a specification from the dataset of open problems instead of a specification-circuit pair from the dataset $D$. We then proceed in the same way as in expert iteration. However, if none of the model predictions satisfy the specification, we do not have a circuit for the next training update. In that case, we continue without a training update and begin sampling either an open problem with probability $p_{open}$ or a labeled specification-circuit pair with probability $1 - p_{open}$ again.

We give the full description for iterating on open problems in Algorithm 2. In contrast to Algorithm 1 on reinforcing learned semantics, we additionally require a dataset of open problems and a probability to select from that dataset.

---

**Algorithm 2** Iterating on Open Problems

---

**Require:** $D_{train}, D_{open}, p_{open}$
1: $M_0 \leftarrow \text{IMITATIONLEARNING}(D_{train})$
2: **for** $i \leftarrow 1$ **to** $n$ **do**
3:     **if** $random() > p_{open}$
4:         Sample $(\varphi, C) \sim D_{train}$
5:         $\{\hat{C}_1, \ldots, \hat{C}_k\} \leftarrow \text{SEARCH}(M_{i-1}, \varphi)$
6:         $C^* \leftarrow \text{VERIFYANDSELECT}(\varphi, \{\hat{C}_1, \ldots, \hat{C}_k\} \cup \{C\})$
7:         $M_i \leftarrow \text{TRAINUPDATE}(M_{i-1}, (\varphi, C^*))$
8:     **else**
9:         Sample $\varphi \sim D_{open}$
10:        $\{\hat{C}_1, \ldots, \hat{C}_k\} \leftarrow \text{SEARCH}(M_{i-1}, \varphi)$
11:        **if** $C^* \in \{\hat{C}_1, \ldots, \hat{C}_k\}$ such that $C^* \models \varphi$
12:           $M_i \leftarrow \text{TRAINUPDATE}(M_{i-1}, (\varphi, C^*))$

---

## 5 EXPERIMENTS

In this section, we report on the experimental results for our new learning approach for reactive synthesis. We begin by first describing the experimental setup, including datasets, architectures, and implementation in Section 5.1. The different variants of our method are then evaluated in Sections 5.2, 5.3, and 5.4 respectively.

### 5.1 EXPERIMENTAL SETUP

We closely follow the experimental setup of previous work to evaluate our method. In particular, we use the same datasets for training and testing as Schmitt et al. (2021). The training dataset comprises 200 000 specification-circuit pairs resulting from a data generation method based on specification patterns and was created with the synthesis tool Strix (Meyer et al., 2018). The trained models are evaluated on three datasets: The holdout dataset created with the same data generation method as the training data is referred to as `Testset`. The `SYNTCOMP` dataset is a collection of challenging benchmarks, containing 145 instances directly taken from the annual reactive synthesis competition. The `Timeouts` dataset was collected during data generation and contains specifications that the synthesis tool Strix could not solve within 120 seconds. For all datasets, we report $pass@k$ rates denoting whether one of the $k$ circuits generated by the model satisfies the specification. Verification is performed with the nuXmv model checker (Cavada et al., 2014).

We evaluate the two architectures that have been previously employed for learning the reactive synthesis problems, hierarchical transformers (Li et al., 2021) and CodeT5 (Wang et al., 2021). A hierarchical transformer architecture was chosen in previous work for its permutation invariance with respect to different orderings of assumption and guarantee properties in the temporal logic specification. We chose the same hyperparameters for the architecture as in previous work, training models with 8 layers, 4 attention heads, an embedding dimension of 256, and a feed-forward neural network dimension of 1024. Token positions of LTL properties are encoded with a tree positional encoding (Shiv & Quirk, 2019). All models are trained with the Adam optimizer (Kingma & Ba, 2015) with betas set to $\beta_1 = 0.9$ and $\beta_2 = 0.98$. We implemented the standard transformer learning rate schedule proposed by Vaswani et al. (2017). The batch size is set to 256. From the CodeT5 model family, we chose the small version, which has about 60 million trainable parameters. We train CodeT5 models with the AdamW optimizer (Loshchilov & Hutter, 2019) with a start learning rate of 0.0005 and a linear learning rate decay. The weight decay is set to 0.1 and the batch size is set to 128.

We implemented all experiments in PyTorch (Paszke et al., 2019) and included the Hugging-Face (Wolf et al., 2020) transformers library for the CodeT5 experiments. We train on a NVIDIA DGX A100 system. All training runs are between 8 and 96 hours. Further training details and parameters are provided in the respective sections.

### 5.2 REINFORCING LEARNED SEMANTICS

We begin by evaluating the semantic reinforcement for both hierarchical transformer models trained from scratch and fine-tuned CodeT5 models. In both cases, we obtain the top prediction of the model using greedy decoding. All results are summarized in Table 1 and further discussed below.

**Hierarchical Transformer (HT).**   In Table 1, we directly compare the supervised learning of the hierarchical transformer with our semantic reinforcement approach in the rows specifying model HT. Specifically, we compare training for 30 000 steps of supervised learning with only 10 000 steps of supervised learning followed by 2 000 steps of semantic reinforcement. We see the pass rates increase for all datasets, with the difference being most pronounced for the $pass@1$ rates.

The training takes about 1 hour and 45 minutes for 10 000 steps of supervised learning and 8 hours and 20 minutes for 2 000 steps of semantic reinforcement when running 4 model checker instances in parallel. While a semantic reinforcement step is more expensive, a relatively small number of steps toward the end of training are sufficient to achieve the performance gains. In Figure 3, we visualize the diminishing returns of changing from supervised learning to semantic reinforcement early in the training process. Further results for different combinations of supervised learning steps and semantic reinforcement steps are shown in Table 5 in Appendix A.

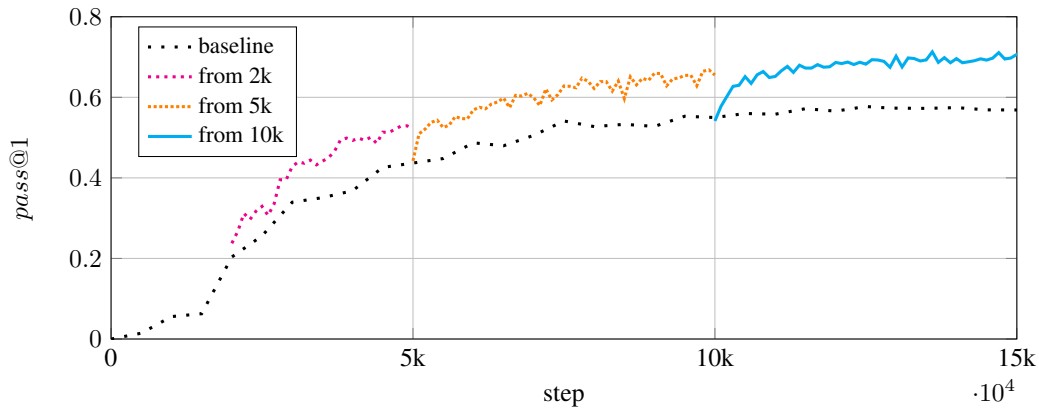

Figure 3: The $pass@1$ rate on validation data over the course of the training comparing supervised learning (baseline), and starting reinforcing learned semantics from step 2 000, 5 000, and 10 000.

Table 1: Pass rates on evaluation datasets for both hierarchical transformer (HT) and CodeT5 showing results for 2 000 steps of reinforcing learned semantics.

| Dataset | Model | Method | pass@1 | pass@4 | pass@8 | pass@16 |
|---|---|---|---|---|---|---|
| Testset | HT | Supervised Learning | 53.6 | 70.4 | 75.8 | 79.9 |
| | | + Semantic Reinforcement | **70.4** | **80.0** | **82.6** | **85.3** |
| | CodeT5 | Fine-tuning | 61.0 | 76.3 | 81.2 | 84.5 |
| | | + Semantic Reinforcement | **70.1** | **81.7** | **85.2** | **87.4** |
| SYNTCOMP | HT | Supervised Learning | 51.9 | 60.0 | 63.6 | 66.8 |
| | | + Semantic Reinforcement | **53.8** | **62.1** | **65.5** | **66.9** |
| | CodeT5 | Fine-tuning | **55.2** | 63.2 | 65.7 | 68.3 |
| | | + Semantic Reinforcement | 53.1 | **63.7** | **67.4** | **71.9** |
| Timeouts | HT | Supervised Learning | 11.7 | 21.1 | 25.9 | 30.1 |
| | | + Semantic Reinforcement | **24.0** | **31.7** | **34.2** | **36.5** |
| | CodeT5 | Fine-tuning | 13.8 | 24.1 | 30.2 | 35.2 |
| | | + Semantic Reinforcement | **27.7** | **36.1** | **39.0** | **42.4** |

We performed an ablation study to investigate how much performance improvement is due to reinforcing correct circuits. In the ablation study, we compare our setting with skipping over correct circuits and only training on specification-circuit pairs that are not solved yet. Note that this resembles hard negative mining techniques employed, for example, in computer vision domains (Shrivastava et al., 2016). The ablation study can be found in Appendix A.2 and concludes that simply not training on already correct circuits behaves similarly to the regular supervised learning setting. We can therefore attribute performance improvements to the reinforcement of correct circuits.

**Pre-trained Code Generation Models (CodeT5).**   In previous work, it was reported that instead of training hierarchical transformers from scratch, fine-tuning pre-trained code generation models such as CodeT5 (Wang et al., 2021) yields comparable or better results for reactive synthesis. We repeat the experiments for hierarchical transformers with CodeT5. The rows specifying model

CodeT5 in Table 1 compare CodeT5 fine-tuned for 30 000 steps with CodeT5 fine-tuned for 20 000 steps and performing an additional 2 000 steps of reinforcing learned semantics. The training takes about 7 hours and 20 minutes for 20 000 steps of fine-tuning and 4 hours and 10 minutes for 2 000 steps of semantic reinforcement. Similar to training transformers from scratch, reinforcing learned semantics improves fine-tuned code models for almost all pass rates on all evaluation datasets. We note that CodeT5's general advantage over hierarchical transformers carries over to the setup with reinforcing learned semantics. More results for different combinations of training steps can be found in Table 12 in Appendix B.

## 5.3 Expert Iteration

In this section, we evaluate the generalization of our method to expert iteration. We obtain the top-k circuit predictions from our model with a beam search (Sutskever et al., 2014) and distinguish two criteria to choose our training target among them. First, we present results for selecting the first circuit that satisfies the specification, and second, the results for choosing the smallest circuit among the ones satisfying the specification. In both cases, we fall back to the circuit from our dataset if no circuit satisfies the specification. We only report results for CodeT5 since CodeT5 outperformed hierarchical Transformers in most evaluations. The results for hierarchical Transformers can be found in Table 7 in Appendix A.3.

Table 2: Pass rates for CodeT5 on evaluation datasets after 20 000 steps of supervised learning and 2 000 steps of expert iteration compared for different beam sizes. Results are averaged over 3 runs.

| Dataset | Beam Size | pass@1 | pass@4 | pass@8 | pass@16 |
|---------|-----------|--------|--------|--------|---------|
| Testset | 1 | 70.1 | 81.7 | 85.2 | 87.4 |
|         | 4 | **74.8** | **84.4** | **87.2** | **89.3** |
| SYNTCOMP | 1 | 53.1 | 63.7 | 67.4 | 71.9 |
|          | 4 | **56.1** | **67.8** | **69.9** | **72.4** |
| Timeouts | 1 | 27.7 | 36.1 | 39.0 | 42.4 |
|          | 4 | **32.1** | **41.3** | **43.9** | **46.5** |

We summarize the results for the first criterion in Table 2, comparing a beam size of 1 and a beam size of 4. Note that a beam size of 1 corresponds to greedy decoding and therefore to the results of the method presented in Section 5.2. We can see through all $pass@k$ rates a clear improvement when moving from Beam Size 1 to 4. For the Testset, the result for $pass@16$ constitutes the best performance in the paper and establishes new state-of-the-art results on the dataset. We note that the improvements come at a higher computational cost, as more model checking calls are made each training step. The number of model checking calls scales linearly in the beam size.

In a second step, we adapt our expert iteration method to optimize for smaller circuits. We optimize our method by selecting the smallest correct circuit from the set of the top-k predictions as the next training target. Notably, the syntactic accuracy during expert iteration with selecting smaller circuits drops from $36\%$ to $0\%$, while the semantic accuracy is still improving (see Figure 4 in the Appendix). This shows that the model's predictions shift strongly away from the original training data, while still being correct.

We show that optimizing for size does not impede performance (see A.4) while generating smaller circuits. We evaluate on the SYNTCOMP dataset, and compare circuit sizes to Strix, the algorithmic state-of-the-art tool in Reactive Synthesis. As shown in Table 3, our circuits are $54\%$ smaller on average, while fine-tuning without Expert Iteration creates circuits that are $46\%$ smaller than Strix's circuits. Expert Iteration improves the circuit size over fine-tuning by $12.5\%$. Absolute results, including for the hierarchical Transformer, can be found in Appendix A.4.

Table 3: Improvement in circuit size over different baselines (percent). Evaluated on SYNTCOMP.

| Improvement of | eval@1 | eval@4 | eval@8 | eval@16 |
|---|---|---|---|---|
| Fine-tuning over Strix | 46.0 | 46.1 | 46.4 | 48.7 |
| Expert Iteration over Strix | 54.6 | 55.2 | 56.5 | 55.0 |
| Expert Iteration over Fine-tuning | 12.5 | 9.1 | 5.7 | 4.7 |

## 5.4 ITERATING ON OPEN SYNTHESIS PROBLEMS

By applying deep learning to reactive synthesis, we aim to enable compute solutions for synthesis problems that existing synthesis tools cannot solve. The Timeouts dataset that we evaluated on in previous sections provides exactly these kind of specifications. It contains specifications that the Strix could not solve within 120 seconds. Note that increasing this timeout does not increase the number of solved instances substantially because of the high complexity of the problem. In the following, we use the dataset to evaluate our method to iterate on open problems. We follow the same setup as in Section 5.3, fine-tuning CodeT5 models for 20 000 steps and then performing 2 000 steps of expert iteration with a beam size of 4. Within the 2 000 steps of expert iteration, we pick a problem of the Timeouts dataset with probability $p_{timeout}$ instead of a problem from the regular dataset. In Table 4 we report the results of the experiment comparing $p_{timeout}$ values of 0.0, 0.25, and 0.5. The results show that we can effectively bootstrap on open synthesis problems and solve more than half of the problems in the dataset that the symbolic synthesis tool was unable to solve. The results for $pass@16$ establish a new state-of-the-art result on the dataset. On other datasets, iterating on timeouts does not clearly improve or decrease performance as shown in the full results Table 14 in Appendix B.3.

Table 4: Pass rates for CodeT5 on holdout portion of Timeouts dataset after 20 000 steps of fine-tuning and 2 000 steps of expert iteration with beam size 4 comparing different probabilities $p_{timeout}$ of including samples from Timeouts during training. Results are averaged over 3 runs.

| Dataset | $p_{timeout}$ | pass@1 | pass@4 | pass@8 | pass@16 |
|---|---|---|---|---|---|
| | 0.0 | 32.1 | 41.3 | 43.9 | 46.5 |
| Timeouts | 0.25 | 38.0 | 45.6 | 48.4 | 50.7 |
| | 0.5 | **40.1** | **47.4** | **49.9** | **51.9** |

## 6 CONCLUSION

We presented a combination of supervised learning and reinforcement learning for learning reactive synthesis. By rectifying the training objective to synthesizing correct circuits rather than imitating a synthesis tool, we showed substantial performance gains over pure supervised learning approaches on both in-distribution and out-of-distribution benchmarks. The tight integration with a model checker makes our approach flexible and allows to include search and other optimization criteria, such as size, in the learning process. Indeed, we showed the ability to discover smaller circuits than algorithmic synthesis tools.

The results of the paper show the potential of deep learning solutions to push the barriers of reactive synthesis. By bootstrapping on open synthesis problems, our approach is capable of solving more than half of the problems that modern synthesis tools cannot solve. The ability to self-improve and progress on a distribution of open problems is particularly interesting synthesis-style problems, and will be an essential step for solving reactive synthesis problems in practice.

## 7 REPRODUCIBILITY STATEMENT

All datasets and libraries used in this work are open source. In Section 5.1 we describe the hyper-parameters for all architectures and training algorithms needed to reproduce the results. We will make our implementation publicly available after the double-blind review period ends. We averaged experimental results over 3 runs to ensure reproducibility.

## 8 LLM USE STATEMENT

In this paper, large language models were used for small refinements when writing the paper and as an assistant tool when implementing the experiments.

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

## A    FULL HIERARCHICAL TRANSFORMER RESULTS

### A.1    SEMANTIC REINFORCEMENT

Table 5: Pass rates across datasets, highlighting the effect of varying the number of Expert Iteration (EI) steps and Supervised Learning (SL) steps.

| Dataset | # SL Steps | # EI Steps | pass@1 | pass@4 | pass@8 | pass@16 |
|---|---|---|---|---|---|---|
| Testset | 10k | 2k | 70.4 | 80.0 | 82.6 | 85.3 |
| | 10k | 5k | 68.7 | 79.3 | 82.2 | 85.6 |
| | 15k | 2k | 67.3 | 78.6 | 83.2 | 86.6 |
| | 15k | 5k | 69.1 | 79.9 | 82.9 | 85.5 |
| SYNTCOMP | 10k | 2k | 53.8 | 62.1 | 65.5 | 66.9 |
| | 10k | 5k | 53.8 | 64.8 | 67.6 | 70.3 |
| | 15k | 2k | 57.2 | 63.4 | 68.3 | 68.3 |
| | 15k | 5k | 57.9 | 68.3 | 71.0 | 71.7 |
| Timeouts | 10k | 2k | 24.0 | 31.7 | 34.2 | 36.5 |
| | 10k | 5k | 25.2 | 32.1 | 34.4 | 36.9 |
| | 15k | 2k | 22.3 | 29.0 | 31.5 | 34.1 |
| | 15k | 5k | 24.5 | 31.5 | 33.8 | 35.6 |

## A.2 HARD NEGATIVE MINING

Table 6: Results for a hierarchical transformer trained for 10 000 steps on the full dataset and for 2 000 steps on the subset of specification–circuit pairs it could not solve up to that point.

| Dataset | pass@1 | pass@4 | pass@8 | pass@16 |
|---------|--------|--------|--------|---------|
| Testset | 55.2 | 72.9 | 78.1 | 82.5 |
| SYNTCOMP | 51.7 | 63.4 | 67.6 | 70.3 |
| Timeouts | 12.7 | 22.6 | 26.7 | 32.3 |

## A.3 EXPERT ITERATION

Table 7: Pass rates across datasets, showing the effect of varying the beam search size alongside the number of Expert Iteration (EI) steps and Supervised Learning (SL) steps.

| Dataset | # SL Steps | # EI Steps | Beam Size | pass@1 | pass@4 | pass@8 | pass@16 |
|---------|-----------|-----------|-----------|--------|--------|--------|---------|
| Testset | 10k | 2k | 1 | 70.4 | 80.0 | 82.6 | 85.3 |
|  | 5k | 2k | 4 | 66.1 | 78.5 | 82.0 | 85.0 |
|  | 10k | 2k | 4 | 75.0 | 84.4 | 87.2 | 89.2 |
| SYNTCOMP | 10k | 2k | 1 | 53.8 | 62.1 | 65.5 | 66.9 |
|  | 5k | 2k | 4 | 41.1 | 56.6 | 63.4 | 65.5 |
|  | 10k | 2k | 4 | 57.2 | 66.9 | 69.0 | 71.7 |
| Timeouts | 10k | 2k | 1 | 24.0 | 31.7 | 34.2 | 36.5 |
|  | 5k | 2k | 4 | 24.5 | 35.4 | 38.8 | 41.4 |
|  | 10k | 2k | 4 | 29.9 | 39.4 | 42.1 | 44.9 |

## A.4 CIRCUIT MINIMIZATION RESULTS

Table 8: Evaluation of the hierarchical transformer models optimized for size vs. optimized for correctness

| Dataset | Criterion | pass@1 | pass@4 | pass@8 | pass@16 |
|---------|-----------|--------|--------|--------|---------|
| Testset | Correctness | 76.4 | 84.0 | 86.7 | 88.8 |
|  | Size | 76.5 | 83.8 | 86.9 | 89.1 |
| SYNTCOMP | Correctness | 59.8 | 66.9 | 69.9 | 72.6 |
|  | Size | 57.2 | 62.8 | 68.0 | 71.0 |
| Timeouts | Correctness | 31.5 | 39.9 | 42.9 | 45.5 |
|  | Size | 31.9 | 39.8 | 43.1 | 45.7 |

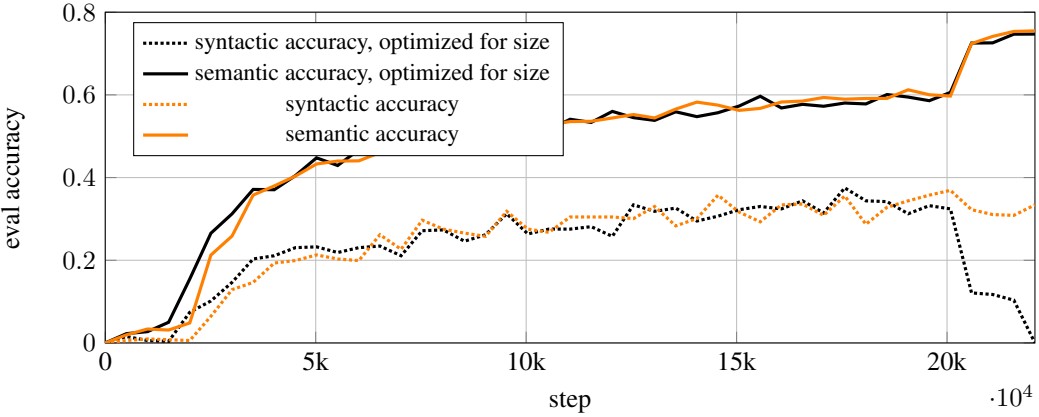

Figure 4: Comparison of semantic and syntactic accuracy during training for Expert Iteration, with and without size optimization.

Table 9: Evaluation of the CodeT5 models optimized for size vs. optimized for correctness

| Dataset | Criterion | pass@1 | pass@4 | pass@8 | pass@16 |
|---|---|---|---|---|---|
| Testset | Correctness | 74.8 | 84.4 | 87.2 | 89.3 |
| | Size | 75.0 | 82.7 | 85.9 | 88.4 |
| SYNTCOMP | Correctness | 56.1 | 67.8 | 69.1 | 72.4 |
| | Size | 55.4 | 59.3 | 64.4 | 68.3 |
| Timeouts | Correctness | 32.1 | 41.3 | 43.9 | 46.5 |
| | Size | 31.6 | 38.3 | 41.8 | 44.9 |

Table 10: Comparison of circuit sizes (gates and latches) on CodeT5 for the SYNTCOMP benchmark, restricted to samples solved by both methods

| Method | eval@1 | eval@4 | eval@8 | eval@16 |
|---|---|---|---|---|
| CodeT5 Fine-tuning | 5.13 | 4.53 | 4.4 | 4.29 |
| + Expert Iteration | 4.49 | 4.12 | 4.15 | 4.09 |
| | | | | |
| HT Training | 4.4 | 4.48 | 4.53 | 4.27 |
| + Expert Iteration | 3.98 | 4.2 | 4.11 | 4.14 |

Table 11: Circuit sizes (gates and latches) on CodeT5 for the SYNTCOMP benchmark, restricted to samples solved by both our method and Strix.

| Method | eval@1 | eval@4 | eval@8 | eval@16 |
|---|---|---|---|---|
| CodeT5 Fine-tuning | 5.36 | 4.99 | 5.2 | 4.83 |
| Strix | 9.93 | 9.25 | 9.7 | 9.41 |
| Expert Iteration on CodeT5 | 4.63 | 4.39 | 4.16 | 4.12 |
| Strix | 10.2 | 9.81 | 9.57 | 9.15 |
| HT Training | 4.71 | 4.76 | 4.86 | 4.53 |
| Strix | 9.45 | 9.51 | 9.32 | 9.12 |
| Expert Iteration on HT | 4.53 | 4.4 | 4.64 | 4.59 |
| Strix | 9.16 | 9.59 | 9.65 | 9.2 |

# B  FULL CODET5 RESULTS

## B.1  SEMANTIC REINFORCEMENT

Table 12: Accuracy of pre-trained CodeT5 across varying numbers of Supervised Learning (SL), Expert Iteration (EI) steps and beam sizes.

| Dataset | SL Steps | EI Steps | pass@1 | pass@4 | pass@8 | pass@16 |
|---------|----------|----------|--------|--------|--------|---------|
| Testset | 30k | 0 | 61.2 | 75.5 | 80.7 | 83.9 |
| | 50k | 0 | 65.9 | 74.6 | 81.5 | 84.8 |
| | 10k | 2k | 66.4 | 78.7 | 83.0 | 85.9 |
| | 10k | 5k | 70.1 | 80.0 | 83.4 | 86.7 |
| | 20k | 2k | 67.2 | 82.3 | 85.5 | 87.3 |
| | 20k | 5k | 74.2 | 84.3 | 87.5 | 89.4 |
| | 30k | 2k | 70.4 | 82.8 | 86.2 | 88.4 |
| | 30k | 5k | 71.7 | 81.4 | 86.4 | 88.3 |
| SYNTCOMP | 30k | 0 | 59.3 | 62.8 | 66.2 | 68.3 |
| | 50k | 0 | 57.9 | 64.1 | 67.6 | 70.3 |
| | 10k | 2k | 43.4 | 58.6 | 62.1 | 66.2 |
| | 10k | 5k | 50.3 | 62.1 | 65.5 | 66.2 |
| | 20k | 2k | 51.7 | 64.1 | 67.6 | 71.0 |
| | 20k | 5k | 57.2 | 64.1 | 65.5 | 69.0 |
| | 30k | 2k | 56.5 | 64.8 | 70.3 | 75.9 |
| | 30k | 5k | 56.5 | 63.4 | 65.5 | 71.7 |
| Timeouts | 30k | 0 | 13.7 | 24.2 | 30.4 | 36.3 |
| | 10k | 2k | 25.8 | 35.4 | 38.6 | 41.0 |
| | 10k | 5k | 29.7 | 38.6 | 40.5 | 43.1 |
| | 20k | 2k | 27.3 | 35.6 | 38.4 | 42.0 |
| | 20k | 5k | 30.2 | 37.7 | 40.4 | 43.7 |
| | 30k | 2k | 25.3 | 34.6 | 38.3 | 41.3 |
| | 30k | 5k | 29.0 | 37.2 | 40.1 | 42.4 |

## B.2  EXPERT ITERATION

Table 13: CodeT5 circuit minimization.

| Dataset | Criterion | pass@1 | pass@4 | pass@8 | pass@16 |
|---------|-----------|--------|--------|--------|---------|
| Testset | Correctness | 76.0 | 85.1 | 87.6 | 89.6 |
| | Size | 75.0 | 82.7 | 85.9 | 88.4 |
| SYNTCOMP | Correctness | 55.6 | 67.1 | 70.8 | 73.8 |
| | Size | 55.4 | 59.3 | 64.4 | 68.3 |
| Timeouts | Correctness | 32.8 | 41.4 | 44.0 | 46.5 |
| | Size | 31.6 | 38.3 | 41.8 | 44.9 |

## B.3 ITERATING ON OPEN PROBLEMS

Table 14: Pass rates for CodeT5 on evaluation datasets after $20\,000$ steps of fine-tuning and $2\,000$ steps of expert iteration with beam size 4 comparing different probabilities $p_{timeout}$ of including timeouts. Results are averaged over 3 runs.

| Dataset | $p_{timeout}$ | pass@1 | pass@4 | pass@8 | pass@16 |
|---------|---------------|--------|--------|--------|---------|
| | 0.0 | 74.8 | 84.4 | 87.2 | 89.3 |
| Testset | 0.25 | **75.2** | 84.3 | **87.6** | **89.6** |
| | 0.5 | 74.5 | **84.8** | 87.4 | 89.3 |
| | 0.0 | **56.1** | 67.8 | 69.9 | 72.4 |
| SYNTCOMP | 0.25 | 55.0 | 68.1 | **71.2** | **73.8** |
| | 0.5 | 53.5 | **68.3** | 70.8 | 71.5 |
| | 0.0 | 32.1 | 41.3 | 43.9 | 46.5 |
| Timeouts | 0.25 | 38.0 | 45.6 | 48.4 | 50.7 |
| | 0.5 | **40.1** | **47.4** | **49.9** | **51.9** |

## C EXPERT ITERATION ALGORITHM

---
**Algorithm 3** Expert Iteration

---
**Require:** $D = \{(\varphi_1, C_1), \ldots, (\varphi_n, C_n)\}$
1: $M_0 \leftarrow \text{IMITATIONLEARNING}(D)$
2: **for** $i \leftarrow 1$ **to** $n$ **do**
3:      Sample $(\varphi, C) \sim D$
4:      $\{\hat{C}_1, \ldots, \hat{C}_k\} \leftarrow \text{SEARCH}(M_{i-1}, \varphi)$
5:      $C^* \leftarrow \text{VERIFYANDSELECT}(\varphi, \{\hat{C}_1, \ldots, \hat{C}_k\} \cup \{C\})$
6:      $M_i \leftarrow \text{TRAINUPDATE}(M_{i-1}, (\varphi, C^*))$

---

