# OpenReview forum: "Learning Reactive Synthesis from Model Checking Feedback"
_ICLR.cc/2026/Conference — Submitted to ICLR 2026_

### Official Review · Reviewer_SnDa · 2025-10-31

**Soundness:** 2
**Presentation:** 2
**Contribution:** 2
**Rating:** 4
**Confidence:** 3

**Summary:**

This paper proposed a new reinforcement learning framework of synthesizing hardware circuits based on the feedback from model checking results.
The experiments are based on open datasets and the results are outperform supervised learning baselines.

Pros:
1. The integration of model checking results and circuit synthesis is interesting.

Cons:
1. Using feedback from formal methods for learning is not novel, the novelty of the method is limited.
2. The experiment results are limited and not convincing.

**Strengths:**

1. I think it is interesting to apply feedbacks from model checking to circuit synthesis, and this integration is quite novel, at least I have never seen it before.
2. Overall the paper is well organized and it is not hard to follow.
3. The literature review is comprehensive.

**Weaknesses:**

1. My major concern of this work is the novelty of using model checking/formal verification feedback for learning, there are many of these works since 2023 and I'm not sure what novel or unique challenges does the circuit synthesis problem introduce. In fact, I have a feeling that this problem is well formulated and easy to solve compare to other problems. Maybe the authors can better elaborate the challenges and why it can be solved by this way.
2. My second concern is the experiments, seems like codeT5 outperforms the HT method, then why we want to have it?
3. I think the authors should clarify how the supervised part work from the problem level, currently, this part is missing in the paper.

**Questions:**

I have a basic question for choosing LTL, why not other logics?
For example, signal temporal logic (STL) can provide dense and continuous feedbacks from the specification satisfaction, why you choose LTL?

---

> ### Author Response · Authors · 2025-11-26
> **Reply to Reviewer SnDa**
>
> We thank the reviewer for their review and for highlighting the novel reinforcement learning framework for circuit synthesis. In the following, we address the weaknesses and questions raised by the reviewer.
>
> "In fact, I have a feeling that this problem is well formulated and easy to solve compare to other problems. Maybe the authors can better elaborate the challenges and why it can be solved by this way."
>
> We want to stress that reactive synthesis problems are not easy to solve. In the case of LTL synthesis, the double-exponential complexity is a major barrier to solving industrial-grade problems and hinders adoption. We believe that deep learning approaches have the potential to make reactive synthesis more applicable, and that the integration of reinforcement learning (which has not been explored in this context) is key. Reinforcement learning enables direct optimization for circuit size, which translates to cost in the context of hardware synthesis. More importantly, it enables us to move beyond the abilities of existing synthesis tools as demonstrated in the paper.
>
> "My second concern is the experiments, seems like codeT5 outperforms the HT method, then why we want to have it?"
>
> We included experiments for both hierarchical transformer and CodeT5 to demonstrate that integrating model checking feedback yields improvements regardless of whether we train neural networks from scratch or fine-tune pre-trained models. Moreover, the hierarchical transformer results provide valuable insights for resource-constrained settings, as these models have only 15 million trainable parameters compared to CodeT5's 60 million.
>
> "I have a basic question for choosing LTL, why not other logics? For example, signal temporal logic (STL) can provide dense and continuous feedbacks from the specification satisfaction, why you choose LTL?"
>
> We chose LTL because it is one of the most influential temporal logics in computer-aided verification. It is the prototype logic for many other logics as well as industrial-grade specification languages such as PSL. Note that STL can also be seen as extending the ideas of LTL to real-time domains. One testament to the foundational nature of LTL is that the annual reactive synthesis competition focuses on LTL synthesis. While the reviewer's proposal of using dense and continuous feedback is interesting, we are not aware of any method for quantifying partial satisfaction of LTL or STL specifications by circuits in a way that would lend itself to optimization.

---

### Official Review · Reviewer_3cFL · 2025-10-31

**Soundness:** 2
**Presentation:** 2
**Contribution:** 2
**Rating:** 2
**Confidence:** 2

**Summary:**

This paper proposes an approach for synthesizing circuits from linear temporal logic (LTL) specifications using machine learning. The method builds on prior work by integrating model checker feedback and adding a search component for circuit size optimization. The approach is evaluated on several datasets.

**Strengths:**

The paper improves the performance of neural circuit-synthesis approaches by integrating model-checker feedback.

**Weaknesses:**

- **W1.** (Minor) The second paragraph of the introduction needs citations.
- **W2.** In my view, the contributions are not very strong. The paper proposes an arguably trivial combination of existing neural circuit-synthesis, feedback-based fine-tuning, and expert-iteration methods.
- **W3.** The differences between the proposed approach and prior approaches are not discussed in the related work section.
- **W4.** Although it is simpler than reactive synthesis, LTL model checking is also computationally hard. Thus, the proposed approach may introduce additional computational burden compared with existing methods, which is neither discussed nor evaluated.
- **W5.** (Minor) The algorithm fragment at the bottom of page 4 is somewhat confusing.
- **W6.** The results for SYNTCOMP in Table 1 are not very impressive.

**Questions:**

What are your thoughts on the weaknesses mentioned above?

---

> ### Author Response · Authors · 2025-11-26
> **Reply to Reviewer 3cFL**
>
> We thank the reviewer for their review and hope to address the reviewer's concerns in the following.
>
> "(Minor) The second paragraph of the introduction needs citations."
>
> We have added citations on previous neural circuit synthesis work and on the limitations of imitation learning.
>
>  "In my view, the contributions are not very strong. The paper proposes an arguably trivial combination of existing neural circuit-synthesis, feedback-based fine-tuning, and expert-iteration methods."
>
> We respectfully disagree with this assessment. While individual methods have been proposed in different contexts, we argue that their combined application to reactive synthesis is both novel and advances the state-of-the-art. Crucially, the combination of supervised learning and reinforcement learning enables minimizing circuits and moving beyond existing synthesis tools, both of which have been demonstrated in the paper.
>
> "The differences between the proposed approach and prior approaches are not discussed in the related work section."
>
> We have discussed the differences from existing solutions in the introduction of the paper and provided details on existing solutions in the second paragraph of Section 3.
>
> "Although it is simpler than reactive synthesis, LTL model checking is also computationally hard. Thus, the proposed approach may introduce additional computational burden compared with existing methods, which is neither discussed nor evaluated."
>
> We concur with the reviewer that this comparison is important and was missing from the paper.
> We have added additional runtime information in the paper and provide it here for convenience. When training a hierarchical transformer with a batch size of 256 for 10,000 steps of supervised learning followed by 2,000 steps of semantic reinforcement, the whole training run takes about 10 hours and 5 minutes. Supervised learning takes roughly 1 hour and 45 minutes, and semantic reinforcement takes 8 hours and 20 minutes. When fine-tuning CodeT5 with a batch size of 128 for 20,000 steps and performing an additional 2,000 steps of semantic reinforcement, the whole training run takes about 11 hours and 30 minutes. Supervised learning takes roughly 7 hours and 20 minutes, and semantic reinforcement takes 4 hours and 10 minutes. While a single step of semantic reinforcement is more expensive, we note that far fewer steps are needed, as analyzed in Section 5.2. With sufficient computational resources, it is possible to further optimize the runtime by running more model checker instances in parallel. In our case, we ran 4 model checker instances in parallel.
>
> "(Minor) The algorithm fragment at the bottom of page 4 is somewhat confusing."
>
> We thank the reviewer for pointing out that the algorithm fragment was not clear. We have removed the fragment and provided the full algorithm in Appendix C.

---

### Official Review · Reviewer_GQr2 · 2025-10-31

**Soundness:** 3
**Presentation:** 3
**Contribution:** 2
**Rating:** 2
**Confidence:** 4

**Summary:**

This paper addresses the limitations of existing deep learning approaches to reactive synthesis—where supervised learning is confined to imitating synthesis tools and reinforcement learning has slow convergence. It proposes a hybrid method that initializes models via supervised learning, then refines them using model checking feedback to prioritize correct circuit synthesis over tool imitation.

Reactive synthesis, which constructs systems satisfying linear temporal logic specifications (critical for hardware design), is computationally hard (2EXPTIME-complete), leading traditional tools to timeout even for small specs. The paper’s hybrid framework first trains an initial model ($M_0$) on 200,000 Strix-generated specification-circuit pairs (supervised phase). In the second phase, it verifies the model’s predicted circuits ($\hat{C}$) with nuXmv: if $\hat{C}$ meets the spec, it reinforces the model with $(\varphi, \hat{C})$; if not, it falls back to the dataset’s correct circuit ($C$).

Three core variants extend the framework: 1) "Reinforcing Learned Semantics" boosts generalization by leveraging correct non-dataset predictions; 2) "Expert Iteration" uses beam search (top-k predictions) to improve performance and minimize circuit size with 54% smaller than Strix on average; 3) "Iterating on Open Problems" samples unsolvable Timeouts dataset to exceed tool capabilities.

Experiments on hierarchical transformers and fine-tuned CodeT5 show state-of-the-art results: CodeT5 with expert iteration hits 89.3%  on Testset and 51.9%  on Timeouts. The method advances reactive synthesis by combining efficiency, correctness, and scalability beyond traditional tools.

**Strengths:**

1.  The paper addresses the limitations of pure supervised or reinforcement learning in reactive synthesis by proposing a two-stage hybrid approach. It initializes models via supervised learning (imitating synthesis tool Strix) but shifts to reinforcing formally verified predictions using model checkers (nuXmv). This refocuses the training objective from "tool imitation" to "generating correct circuits," enabling better generalization.

2. Extending the core framework with expert iteration  delivers both performance and efficiency improvements. With beam size 4, CodeT5 achieves an 89.3%  on the Testset  and reduces circuit size by 54% on average compared to Strix—surpassing the 46% reduction from fine-tuning alone. Critically, optimizing for smaller circuits does not harm correctness, as semantic accuracy remains high even when syntactic alignment with training data drops to 0%.

3. The framework uniquely enables iterative improvement on "open problems" (specs Strix times out on). By sampling these specs during training (with \(p_{timeout}=0.5\)), CodeT5’s performance on the Timeouts dataset reaches 51.9%—solving over half the specs traditional tools cannot.

**Weaknesses:**

1.  The framework depends heavily on Strix for dataset generation and nuXmv for model checking. It does not validate performance with other synthesis tools (e.g., SYNTCOMP competitors) or model checkers (e.g., SPIN), leaving uncertainty about whether results hold across different toolchains.

2. No comparison with other state-of-the-art approaches such as NeuroSnyt. The experimental evaluation cannot reflect the advantage of the proposed approach over these existing approaches and tools without any experimental evaluation.

3.  While expert iteration boosts performance, the paper only notes “linear scaling” of model-checking calls with beam size. It lacks quantitative data on training time/memory costs for large beam sizes (e.g., >4) or complex specs, making it hard to assess feasibility for resource-constrained scenarios.

4.  Though the proposed method solves 51.9% of Timeouts specs, the paper does not analyze if these solved “open problems” share specific patterns. It also fails to test if the method generalizes to open problems from non-Strix-generated datasets, weakening claims of transcending tool limits.

**Questions:**

1. The framework relies on Strix for dataset generation and nuXmv for model checking. Have you tested whether the performance gains  hold when using other synthesis tools (e.g., SYNTCOMP competitors like ltlsynt) or model checkers (e.g., SPIN)? If not, what justifies the exclusive dependence on these two tools for generalizability?

2. When optimizing circuit size via expert iteration, syntactic accuracy drops to 0% while semantic accuracy improves—but the paper does not analyze why this decoupling occurs. Could you explain the key mechanisms that let the model diverge from the dataset’s syntactic patterns yet retain or enhance correctness, and whether this decoupling risks unexpected errors in complex, real-world specs?

3. For open problem iteration, you sample Timeouts specs with \(p_{timeout}=0.5\) to boost performance. Have you explored higher \(p_{timeout}\) values e.g., 0.75 to see if open problem training can further improve pass rates, or does a ceiling exist due to the model’s reliance on initial supervised data from Strix?

---

> ### Author Response · Authors · 2025-11-26
> **Reply to Reviewer GQr2**
>
> We thank the reviewer for their review and hope to address their concerns and questions.
>
> "Have you tested whether the performance gains hold when using other synthesis tools (e.g., SYNTCOMP competitors like ltlsynt) or model checkers (e.g., SPIN)? If not, what justifies the exclusive dependence on these two tools for generalizability?"
>
> Our approach relies on the model checker for a binary correct/incorrect signal. Therefore, using a different model checker would not affect our results. The only effect a different model checker could have is on runtime. Given that nuXmv is a modern and fast solver, and our implementation parallelizes model checker calls, we expect only minor changes in runtime.
> As for the synthesis tool, we relied on Strix for comparison with previous work. Using the same dataset from previous work (synthesized with Strix), we were able to conclude that performance improvements were due to the semantic reinforcement. As the winner of the 2023 synthesis competition, Strix constitutes a modern solver, representative of the capabilities of today's synthesis tools.
>
> "No comparison with other state-of-the-art approaches such as NeuroSnyt. The experimental evaluation cannot reflect the advantage of the proposed approach over these existing approaches and tools without any experimental evaluation."
>
> In Table 1, we directly compare with existing approaches. The rows labeled 'Supervised Learning' and 'Fine-tuning' correspond to existing baseline approaches (training hierarchical transformers and fine-tuning CodeT5). Minor differences from previously reported numbers are due to our replication of the experimental setup. The rows labeled 'Semantic Reinforcement' show our proposed method and directly reflect the advantage over existing methods. Regarding NeuroSynt, we note that it is a portfolio solver combining both neural and symbolic solvers into a single framework. The performance of the neural solver in NeuroSynt is comparable to that of the Supervised Learning baseline. While our approach can directly improve the neural component of such systems, this work focuses on improving pure neural methods rather than hybrid solver integrations.
>
> "While expert iteration boosts performance, the paper only notes "linear scaling" of model-checking calls with beam size. It lacks quantitative data on training time/memory costs for large beam sizes (e.g., >4) or complex specs, making it hard to assess feasibility for resource-constrained scenarios."
>
> We hope to clarify the feasibility for resource-constrained scenarios with additional runtime information in the following. When performing semantic reinforcement or expert iteration, the runtime of a single training step is dominated by the model checking calls. As a concrete example, a single training step of supervised learning for CodeT5 takes about 1.32 seconds. In contrast, a single step of expert iteration with beam size 1 (no beam search) takes about 17.1 seconds. Hence, the runtime increases linearly with the number of beams. For example, with beam size of 4, one step of expert iteration takes about 64.8 seconds. With sufficient computational resources, the runtime can be decreased by performing model checking calls in parallel rather than sequentially. In our case, we ran 4 model checker instances in parallel, thereby decreasing the time for a single step of expert iteration with beam size 4 to 18.0 seconds on average.
>
> "When optimizing circuit size via expert iteration, syntactic accuracy drops to 0% while semantic accuracy improves—but the paper does not analyze why this decoupling occurs. Could you explain the key mechanisms that let the model diverge from the dataset's syntactic patterns yet retain or enhance correctness, and whether this decoupling risks unexpected errors in complex, real-world specs?"
>
> We first note that Strix does not necessarily synthesize minimal circuits; in fact, it often produces circuits larger than the minimal circuit satisfying the specification. By changing the training objective from correctness to circuit size, we incentivize the model to diverge from the original training distribution, which explains the sharp drop in syntactic accuracy. Whether this change in the training objective compromises performance is shown in Table 8 in Appendix A.4. We directly compare optimizing for correctness versus optimizing for size and observe no large differences in performance, including on complex SYNTCOMP benchmarks.
>
> "For open problem iteration, you sample Timeouts specs with (p_{timeout}=0.5) to boost performance. Have you explored higher (p_{timeout}) values, e.g., 0.75, to see if open problem training can further improve pass rates, or does a ceiling exist due to the model's reliance on initial supervised data from Strix?"
>
> While further improvements are possible, they are relatively small, with pass rates improving by roughly one percentage point when moving from p_{timeout}=0.5 to p_{timeout}=0.75.

---

### Official Review · Reviewer_BBPC · 2025-11-03

**Soundness:** 3
**Presentation:** 3
**Contribution:** 2
**Rating:** 4
**Confidence:** 4

**Summary:**

Reactive synthesis is the problem of synthesizing finite-state models from temporal logic specifications. This paper explores if deep learning can be used to solve this problem. Compared to earlier attempts to use ML for reactive synthesis, the new ideas include use of a model checker to give feedback to update the model, use of top-k predictions for improving the quality of learnt solutions, and iterating on problems that model fails to solve. The methods are implemented and evaluated on benchmarks for synthesis competitions.

**Strengths:**

As authors explain, reactive synthesis has a long history of research. This is a computationally challenging problem, and thus there is a clear motivation to explore if deep learning can lead to better techniques. The 3 ideas to improve basic supervised learning are all natural, and likely not previously explored in this context.

**Weaknesses:**

The ideas proposed to improve supervised learning (e.g. using a verifier, model checker in this particular context, for RL feedback) are all standard from ML literature, so there is little conceptual novelty. Yet the paper will be a valuable contribution if the experimental results were strong, but that does not seem to be the case.

**Questions:**

1. Did you try using state-of-the-art LLMs (GPT-5 for instance with some prompting) to solve any of these problems
2. Is there more detailed analysis of computational effort needed to solve these benchmarks
3. Is there some qualitative analysis of some case study that indicates that now we can use reactive synthesis to solve a problem of practical interest

---

> ### Author Response · Authors · 2025-11-26
> **Reply to Reviewer BBPC**
>
> We thank the reviewer for their review and hope to answer the reviewer's questions in the following.
>
> "Did you try using state-of-the-art LLMs (GPT-5 for instance with some prompting) to solve any of these problems"
>
> While this was not the focus of our work, we share the reviewer's curiosity about that topic. In our early experiments, LLMs could not generate satisfying circuits. Reactive synthesis is a highly complex problem, and all methods currently addressing this topic, whether algorithmic or deep learning-based, are highly involved.
>
> "Is there more detailed analysis of computational effort needed to solve these benchmarks"
>
> On average, problems from these benchmarks can be solved within 1-2 seconds both by synthesis tools (on a regular desktop machine) and our CodeT5 models (on an NVIDIA A100 40GB GPU). However, this average case is misleading: for some benchmarks, the doubly exponential complexity quickly becomes an insurmountable barrier, making them infeasible to solve with traditional algorithmic methods. Our results on the timeouts dataset demonstrate that we can make progress on such challenging benchmarks with our deep learning approach.
>
> "Is there some qualitative analysis of some case study that indicates that now we can use reactive synthesis to solve a problem of practical interest"
>
> The practical interest in reactive synthesis has been demonstrated through various case studies in robotics [1], hardware circuits [2], device drivers [3], and  FPGA games [4]. The annual reactive synthesis competition [5] is another testament to the community's interest in the problem. In this work, we focused on the competition since it constitutes the most comprehensive and up-to-date collection of benchmarks.
>
> [1] Hadas Kress-Gazit, Georgios E. Fainekos, George J. Pappas. Temporal-Logic-Based Reactive Mission and Motion Planning. IEEE Transactions on Robotics. 2009.
>
> [2] Yashdeep Godhal, Krishnendu Chatterjee, Thomas A. Henzinger:
> Synthesis of AMBA AHB from Formal Specification: A Case Study. International Journal on Software Tools for Technology Transfer. 2013.
>
> [3] Mona Vij, John Keys, Arun Raghunath, Scott Hahn, Vincent Zimmer, Leonid Ryzhyk,
> Adam Christopher Walker, and Alexander Legg. Device Driver Synthesis. Intel Technology Journal. 2013.
>
> [4] Gideon Geier, Philippe Heim, Felix Klein, Bernd Finkbeiner. Syntroids: Synthesizing a Game for FPGAs using Temporal Logic Specifications. FMCAD 2019.
>
> [5] https://www.syntcomp.org

---

### Meta-Review · Area_Chair_s3pP · 2026-01-06

**Summary:**

This paper proposes a novel strategy for finetuning LLMs, that starts by using supervised learning and then switches to reinforcement learning based on feedback from the model checker.

**Reviewer Concerns:**

There were varying concerns across reviewers; two stood out that were brought up by several reviewers. First, three reviewers had significant concerns about the scope of the contributions. The methodology proposed in the paper is largely based on existing ideas, and it is not clear that it has much novelty. While the authors provided some discussion on this point, I don't believe that it significantly changes the assessment of novelty. Second, two reviewers had concerns about the baselines (one pointed out NeuroSynt, another pointed out GPT-5). Again, I did not find the reviewers' response to be sufficiently convincing, especially in light of the previous concern.

**Reviewer Scores:**

Overall, I think it is unlikely that the response would have moved any of the scores, given both the nature of the main concerns and the fact that the response did not include significant new information compared to the paper.

---

### Decision · Program_Chairs · 2026-01-26

Reject